

# Sexual dimorphism in skull size and shape of *Laticauda colubrina* (Serpentes: Elapidae)

Bartosz Borczyk

Department of Evolutionary Biology and Coservation of Vertebrates, University of Wroclaw, Wrocław, Poland

## ABSTRACT

**Background**. Sexual dimorphism in size and shape is widespread among squamate reptiles. Sex differences in snake skull size and shape are often accompanied by intersexual feeding niche separation. However, allometric trajectories underlying these differences remain largely unstudied in several lineages. The sea krait *Laticauda colubrina* (Serpentes: Elapidae) exhibits very clear sexual dimorphism in body size, with previous studies having reported females to be larger and to have a relatively longer and wider head. The two sexes also differ in feeding habits: males tend to prey in shallow water on muraenid eels, whereas females prey in deeper water on congerid eels.

**Methods**. I investigated sexual dimorphism in skull shape and size as well as the pattern of skull growth, to determine whether males and females follow the same ontogenetic trajectories. I studied skull characteristics and body length in 61 male and female sea kraits.

**Results**. The sexes differ in skull shape. Males and females follow distinct allometric trajectories. Structures associated with feeding performance are female-biased, whereas rostral and orbital regions are male-biased. The two sexes differ in allometric trajectories of feeding-related structures (female biased) that correspond to dietary divergence between the sexes.

**Conclusions**. Sea kraits exhibit clear sexual dimorphism in the skull form that may be explained by intersexual differences in the feeding habits as well as reproductive roles. The overall skull growth pattern resembles the typical pattern observed in other tetrapods.

Corresponding author
Bartosz Borczyk,
bartosz.borczyk@uwr.edu.pl

## INTRODUCTION

Sexual dimorphism results from the interplay between natural and sexual selection (*Darwin, 1871*; *Wallace, 1889*). Male and female morphology can diverge due to different reproductive roles, for example through adaptations to copulation, mate searching, pregnancy *etc*. In addition, there may also be ecological differences between sexes like feeding niche separation or dispersal strategies that result from different reproductive roles or as an adaptation to reduce intraspecific resource competition (*e.g.*, *Hendrick & Temeles, 1989*; *Shine, 1989*; *Vincent & Herrel, 2007*). Moreover, some of the morphological or physiological differences may simply be structural consequences of size dimorphism,

and do not result from sexual selection. From an ontogenetic perspective, these intersexual divergences may be products of increased allometric slopes for given traits in one of the sexes, may be body size independent (increased allometric intercept) or both (*Bonduriansky, 2007*).

The skull is among the most complex skeletal structures. Its two basic functions comprise feeding and protection of the central nervous system and sense organs, but it is also involved in other functions, like mate competition, social signaling, defence and locomotion (see *Cundall & Irish, 2009*). Snake skulls are particularly interesting, since these animals are gape-limited predators that are unable to portion or crush prey (with the exception of a few species of the crab-eating Homalopsinae (*Jayne, Voris & Ng, 2002*) and kukri snakes that eviscerate their prey (*Bringsøe et al., 2020*; *Bringsøe et al., 2021*; *Bringsøe & Holden, 2021*)) as most tetrapods can do. Snakes have adapted to a wide variety of habitats, and include fossorial, ground-dwelling, arboreal and aquatic species. They have also developed several strategies for subduing their prey, and feed on a variety of prey types (*Greene, 1997*). These adaptations have been accompanied by changes in skull form, which shows enormous diversity among snakes (*Cundall & Irish, 2009*). However, although there have been a number of studies dedicated to snake skull diversity (*e.g.*, *Cundall & Irish, 2009*; *Oliveira et al., 2020*; *Rajabizadeh et al., 2021*; *Strong, Palci & Caldwell, 2021*) little is known about the ontogenetic trajectories underlying its intra-specific diversity and intersexual variation at the species level.

Studies on the allometry of snake skulls are restricted to members of a few families. *Monteiro (1998)* and *Jayne et al. (2022)* studied three Boidae species; *Rossman (1980)*, *Young (1989)*, *Hampton (2014)*, *Hampton & Kalmus (2014)* and *Andjelković, Tomović & Ivanović (2016)* studied Natricinae snakes; *Jayne et al. (2022)* studied single Colubrinae species; *Murta-Fonseca & Fernandes (2016)* and *Murta-Fonseca et al. (2019)* focused on Xenodontinae; *Dos Santos & da Costa Prudente (2022)* studied malacophagous Dipsadidae; *Hampton & Moon (2013)* studied a single species of Crotalinae; *Borczyk et al. (2021)*, *Patterson et al. (2022)* and *Ammresh et al. (2023)* studied three species of Elapidae. In addition to the list, *Palci, Lee & Hutchinson (2016)* compared *Anilios bicolor* (Scolecophidia), *Cylindrophis ruffus* (Cylindrophidae), *Aspidites ramsai* (Pythonidae), *Acrochordus arafurae* (Acrochordidae) and *Notechis scutatus* (Elapidae); however, their sample comprises juvenile-adult pairs for each species, not growth series. Some of these studies ignored the potential effect of sexual dimorphism and only *Camilleri & Shine (1990)*, *Murta-Fonseca et al. (2019)* *Borczyk et al. (2021)* and *Dos Santos & da Costa Prudente (2022)* addressed this topic directly, confirming some differences between males and females. However, sexual dimorphism in head size and shape has been reported in a variety of snake species (*e.g.*, *Shine, 1993*; *Vincent & Herrel, 2007*; *Borczyk, 2015*; *Tamagnini et al., 2018*; *Abegg et al., 2020*) and *Faiman et al. (2018)* reported sexual dimorphism in eye size in several snake species. The intersexual divergence of head size and shape is expected in species that show feeding niche separation, *i.e.*, shift from one to another prey type by one of the sexes (*Camilleri & Shine, 1990*).

The sea krait, *Laticauda colubrina*, is a member of one of two independent marine radiations of elapid snakes. It is a highly venomous snake that feeds almost exclusively on

fishes, mainly eels (*Pernetta, 1977*; *Voris & Voris, 1983*; *Shetty & Shine, 2002*; *Shine et al., 2002*; *Heatwole, 1999*). Males and young snakes tend to prey in shallow water on muraenid eels, whereas adult females prey in deeper water on congerid eels (*Pernetta, 1977*; *Shetty & Shine, 2002*; *Shine et al., 2002*). Previous studies have found this species to exhibit very clear sexual dimorphism in body size, with females being bigger and having relatively longer and wider heads (*Shetty & Shine, 2002*; *Shine et al., 2002*). It has also been reported that there are some intersexual differences in skull form (parietal, frontal, nasal, supratemporal, dentary, mandible, and prefrontal bones; *Camilleri & Shine, 1990*). However, the latter authors were more focused on so-called "trophic morphology", did not analyse the allometry and studied a relatively small sample, so some differences might have been undetected (see discussion).

In this article I address the following questions: (1) Do females differ from males in body shape, (2) do male and female sea krait skulls differ in their shape and size, and if so (3) do these differences reflect different allometric growth trajectories of males and females, or is the larger female skull shape a predictable extrapolation along the males' allometric trajectory? Specifically, I expect female-biased sexual differences in the bones involved in prey capture and ingestion as a consequence of feeding niche separation (females eat larger prey), and male-biased dimorphism in the characters supporting the sensory organs as an adaptation to mate searching.

## MATERIALS & METHODS

### Measurements

I measured 60 cleared and dried skulls of *Laticauda colubrina* (33 females, 27 males) from the collection of the Field Museum of Natural History, Chicago (Appendix 1). However, some specimens have missing or broken bones, and in three cases (two females and one male) the measurements could not be taken from some specific distances (*i.e.,* missing or broken both supratemporals in female #FMNH 236269), and these specimens were removed from some analyses. Data on the snout to vent length (SVL) and body weight (BW) were recorded for 38 specimens (20 females and 18 males), and BW was measured prior to fixation. Female specimens ranged from 605 to 1440 mm SVL, and their skull length ranged from 13.14 to 31.57 mm, and male specimens ranged from 587 to 888 mm and 13.31–24.18 mm respectively; thus this covers the series from subadult to adult specimens. The following skull measurements (Fig. 1) were taken with a digital calliper to the nearest 0.1 mm (raw data are in Data S1): skull length (SL) measured from the tip of the premaxillary bone to the end of basioccipital condyle; skull height (SH) measured at the highest point of the braincase; skull width (SW) at the widest point of the braincase; parietal width 1 (PW1) at the postorbital articulation; parietal width 2 (PW2) at the narrowest point of the parietal bone; parietal length (PAR) measured along the midline; nasal component length (NCL) measured from the tip of the premaxillary bone to the caudal end of the nasal, septomaxillary and vomerine bones and frontal articulation; nasal length (NL) measured between the most rostral and most caudal edges of the nasal bones' lamina; nasal width (NW) at the widest point of the nasal lamina; frontal length (FL)

between the most rostral and most caudal edges of the interfrontal midline; frontal width (FW1) at the fronto-parietal suture; frontal width (FW2) at the narrowest point; palatine length (PLL) from the tip to the most caudal point of the articulation process; Pterygoid length (PTL) from the most rostral point of the palato-pterygoid articulation process to the caudal tip of the bone; pterygoid tooth row length (PTTL); width of the premaxillary bone (PMW); length of the retroarticular process (PRETR) from the quadrate articulation to the caudal tip of the bone; length of the maxilla (MXL); length of the mandible (MDL); length of the in-lever of the mandible (MD2L) from the quadrate articulation to the anterior tip of the bone; length of the dentary bone (DENT) measured along its dorsal (tooth bearing) process; mandibular fossa length (FMDB); length of the ectopterygoid bone (ECT); quadrate bone length (QL); quadrate crest length (CQL); prefrontal bone height (PFH); length of the supratemporal bone (STP).

Of these distances the following are feeding related structures: CQL, DENT, ECT, FMDB, MDL, MD2L, MXL, PLL, PRETR, PTL, PTTL, QL and STP. Others, like nasal width and length, prefrontal height, and frontal width, are related to support of the sensory organs (vomeronasal organ and eyes). NCL is involved both in the supporting the chemosensory structures and feeding (rhinokinetism: *Cundall & Shardo, 1995*). Premaxilla width, parietal width and skull length, width and height reflect the overall skull form/shape.

## Statistical analyses

To test for sexual size dimorphism in body length I ran an ANOVA on SVL. The skull length, width and height (SL, SW and SH respectively) roughly correspond to head dimensions. Thus to find if there is sexual dimorphism in overall skull/head size I ran an MANCOVA with SL, SW and SH as dependent variables and SVL as the covariate.

To analyze potential shape differences separately of size I standardized the raw measurement by transforming into the log-shape ratios (*Mosimann & James, 1979*). The log-shape ratio is calculated in the following way: the size of the individual is calculated as geometric mean of all measurements. Then each of the measurements is divided by individual "size" (geometric mean) and the result is the shape ratio which is then log-transformed into log-shape ratio. As such the log shape ratios are size-free shape variables (*Mosimann & James, 1979*; *Claude, 2013*). I ran principal component analysis (PCA) on the log-shape ratio variables using the covariance matrix. PCs with eigenvalues of 1 and higher have been considered and then tested with MANOVA. Variables with highest loadings (>0.7) on PC 1 that was shown to be responsible for between-sex differences (see Results) were further explored with MANOVA. In order to find if the overall skull shape changes with size and if follows the same pathway in both sexes I ran multivariate regression with subsequent ANCOVA on regression score on log-shape ratio variables (*Mosimann & James, 1979*). This has been done using the geomorph package (*Adams & Otarola-Castillo, 2013*; *Baken et al., 2021*; *Adams et al., 2022*) for R v. 4.2.2 (*R Core Team, 2022*).

The allometric coefficients were estimated from the regression of log-transformed skull measurements against the log-transformed skull length (SL) and log-transformed snout to vent length (SVL). These two different baselines were considered because the SVL itself may be sexually dimorphic, and relying only on this size measure may produce skewed

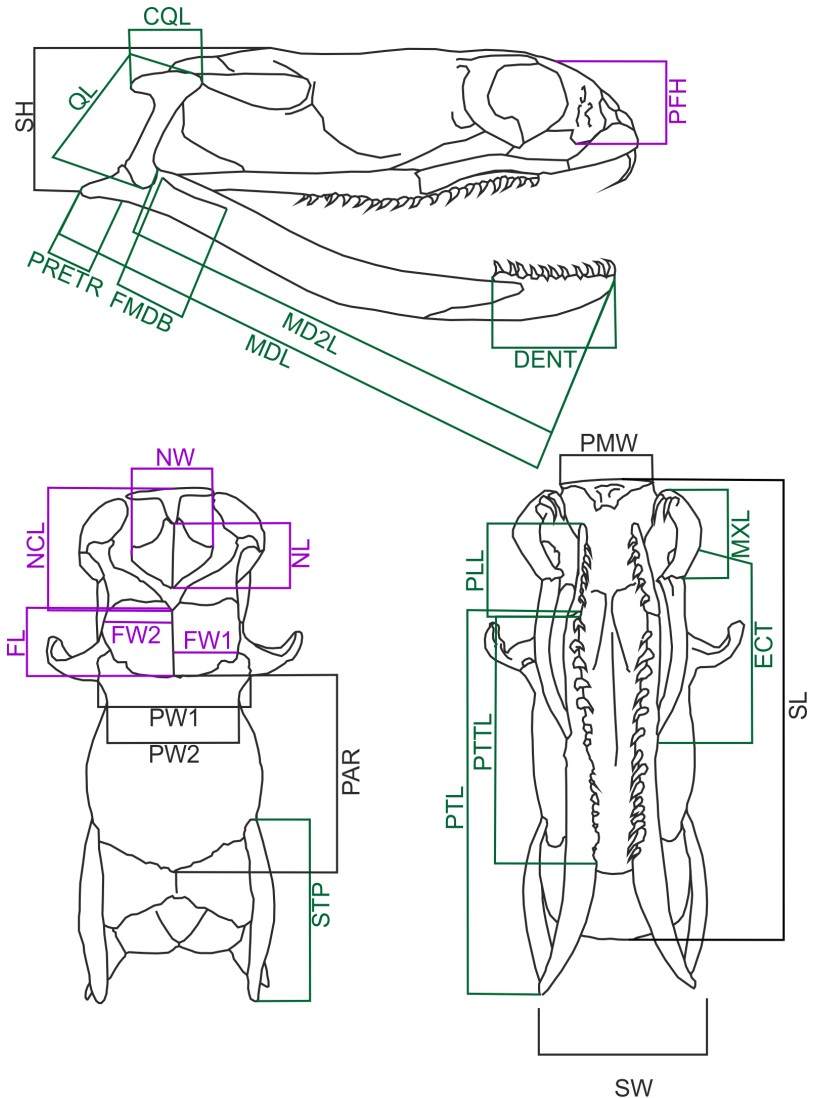

**Figure 1 Diagramatic representation of sea krait (*Laticauda colubrina*) skull.** Skull of *Laticauda colubrina* in lateral (A), dorsal (B) and ventral (C) aspect showing measurements used in the present study. The feeding-related structures are indicated green, and sensory-supporting structures are indicated violet (for further explanation see the main text). Abbreviations: CQL, quadrate crest length; DENT, length of the dentary bone; ECT, length of the ectopterygoid bone; FL, frontal length; FMDB, mandibular fossa length; FW1, frontal width at the fronto-parietal suture; FW2, frontal width at the narrowest point; MD2L, length of the in-lever of the mandible; MDL, length of the mandible; MXL, length of the maxilla; NCL, nasal component length; NL, nasal length; NW, nasal width; PAR, parietal length; PFH, prefrontal bone height; PLL, palatine length; PMW, width of the premaxillary bone; PRETR, length of the retroarticular process; PTL, pterygoid length; PTTL, pterygoid tooth row length; PW1, parietal width 1; PW2, parietal width 2; QL, quadrate bone length; SH, skull height; SL, skull length; STP, length of the supratemporal bone; SW, skull width.

results and interpretation. Because the measurement error was present in both the x- and y-variables the slopes were calculated using Reduced Major Axis Regression (*Sokal & Rohlf, 1995*). Although the Type II Regression Model (or Major Axis Regression) may seem to be

more appropriate in estimating the allometric slopes (see *Warton et al., 2006*), I decided to use RMA Regression as it allows direct comparisons with the similar studies on other species (*Hampton & Moon, 2013*; *Hampton, 2014*; *Hampton & Kalmus, 2014*; *Borczyk et al., 2021*; *Patterson et al., 2022*). All calculations were done using Statistica and software by *Bohonak & van der Linde (2004)*.

## RESULTS

In the studied sample females were longer than males (mean $\pm$ SD 1079.53 $\pm$ 40.56 cm and 783.56 $\pm$ 41.67 cm, ANOVA test: $F_{1,35} = 25.905$, $p = 0.000012$). However, the sexes did not differ in SL, SW or SH when corrected for SVL (MANCOVA test: Wilks' $\lambda = 0.88$, $F = 1.495$, $p = 0.234$).

The PCA on log-shape ratios yielded with 8 PCs, together explaining 77.23% of variation (Table S1, Fig. 2). However, further MANOVA on PC scores showed that male and female sea kraits differ (Wilks' $\lambda = 0.461$, $F_{8,48} = 7.008$, $p < 0.001$) and only the first PC of the eight is responsible for the differences ($F = 45.52$, $p < 0.001$) between the sexes. The variables with loadings on PC1 above $|0.7|$ are SL, PW1, PW2, FL, MD2L, QL and STP (Table 1). MANOVA ran on these log-shape ratios confirmed sexual shape dimorphism in skull form (Wilks' $\lambda = 0.496$, $F_{7,49} = 7.103$, $p < 0.001$). The male biased log-shape ratios are SL, PW1, PW2, and FL, whereas MD2L, QL and STP are female biased (Table 1). Further, multivariate regression revealed male and female skulls of sea kraits follow different allometric slopes and the sexual shape dimorphism does not result from extrapolation of male trajectory onto larger females (Table 2, Fig. 3).

### Allometry

The allometric coefficients for the studied traits are summarized in Figs. 4 and 5 (confidence intervals and determination coefficients for the allometric equations are presented in the Tables S2 and S3). Skull measurements (SL, SH,) scaled with slightly negative allometry with respect to SVL, with the exception of SW, which showed isometry (females) or slightly positive allometry (males). When scaled against SL most of the characteristics showed positive allometry or isometry, and only PW2 showed negative allometry. Table 3 summarizes the allometric coefficient of other snake species from published studies that used homologous skull distances.

## DISCUSSION

### Skull size and shape

Although *Camilleri & Shine (1990)*, *Shetty & Shine (2002)* and *Shine et al. (2002)* previously reported that female sea kraits have relatively longer and wider heads this was not reflected by the overall skull shape and size in the current study; in this case the roughly corresponding traits are MDL (for typical head length) and SW (for head width). Both male and female skulls were uniform and their overall proportions changed in the same way when scaled against both SVL and SL. There may be two explanations for these differences. First, my measurements were not homologous with those of the above mentioned authors. For

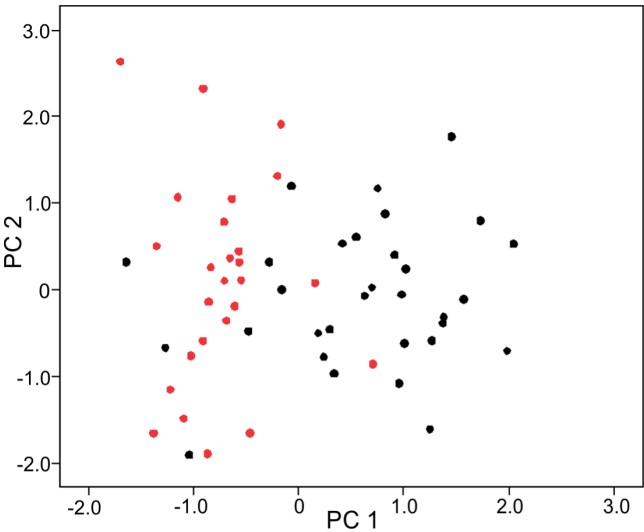

**Figure 2** Principal component (PC) 1 and 2 scatterplot of log-shape ratios of skull measurements of male and female sea kraits, *Laticauda colubrina.* Black dots represent females and red dots represent males.

**Table 1 Results of MANOVA on the log-shape ratios of skull measurements of *Laticauda colubrina*.** Only the variables with PC loadings higher than |0.7| have been used. These variables were log-shape ratios of SL, PW1, PW2, FL, MD2L, QL, PFL and STP.

|  | SS | df | MS | F | p |
|---|---|---|---|---|---|
| lsrSL | 0.039 | 1 | 0.039 | 23.317 | <0.001 |
| lsrPW1 | 0.106 | 1 | 0.106 | 26.330 | <0.001 |
| lsrPW2 | 0.619 | 1 | 0.619 | 39.212 | <0.001 |
| lsrFL | 0.153 | 1 | 0.153 | 25.240 | <0.001 |
| lsrMD2L | 0.018 | 1 | 0.018 | 29.918 | <0.001 |
| lsrQL | 0.137 | 1 | 0.137 | 11.597 | 0.001 |
| lsrSTP | 0.124 | 1 | 0.124 | 25.463 | <0.001 |

**Table 2 ANCOVA results for regression score from multivariate regression on log-shape ratios on log-geometric mean (logGM) of skull measurements of *Laticauda colubrina*.**

|  | df | SS | MS | $r^2$ | F | Z | p |
|---|---|---|---|---|---|---|---|
| logGM | 1 | 2.9718 | 2.97179 | 0.286 | 22.9184 | 4.3797 | 0.001 |
| Sex | 1 | 0.3299 | 0.32987 | 0.032 | 2.544 | 2.9079 | 0.003 |
| logGM:sex | 1 | 0.2336 | 0.2336 | 0.022 | 1.8016 | 1.8897 | 0.026 |
| Residuals | 53 | 6.8724 | 0.12967 | 0.66 |  |  |  |
| Total | 56 | 10.4077 |  |  |  |  |  |

example, the "head width" as measured in previous studies reflects the widest overall point of the head, and thus includes the skull width increased by the quadrate projection and mandible, which extends posteriorly, plus soft tissues (jaw adductor musculature, salivary and venom glands and skin with scales). Jaw adductor musculature increases with positive

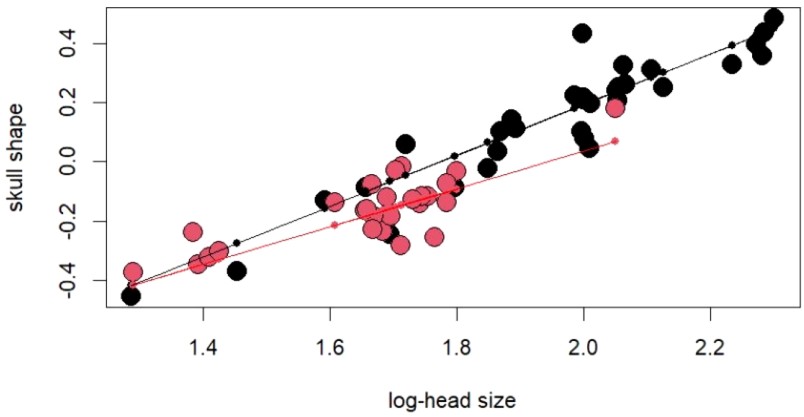

**Figure 3** For regression score from multivariate regression on log-shape ratios on log-head size (log-geometric mean of skull measurements) of *Laticauda colubrina*. Black colour indicates females and red indicates males.

allometry with respect to skull length (*Vincent & Herrel, 2007*), and thus it may contribute to the overall head width dimorphism. In contrast, my skull measurements were taken at well-defined points (*e.g.*, at bone sutures), and were not necessarily in the same plane as the widest head point and of course, did not cover the soft tissues.

Although females and males did not differ in overall skull shape (as reflected by SL, SH and SW), specific bones within the skull did show significant sexual dimorphism. The sexually dimorphic differences that are female-biased are those related to feeding efficiency, which may be attributed to sexual and ontogenetic differences in feeding behaviour (from muraenid eels in males and young snakes to larger congerid eels in females). However, *Camilleri & Shine (1990)* found no sexual differences in the so-called "trophic structures" of sea kraits, and asserted that "sea snakes did not show significant differences in the lower jaw although observed differences were in the predicted direction". A possible explanation for this discrepancy is the relatively small sample size in *Camilleri & Shine (1990)*, comprising 13 males and 13 females, which may have led to type II error.

Male-biased dimorphism in head shape is restricted to the frontal (FL) and parietal (PW1 and PW2) regions. These are the skull parts that contain two of the major sensory structures (olfactory bulbs and eyes). Pheromone-driven mate searching is well documented in snakes (*Mason & Parker, 2010*) and may explain male-biased dimorphism in the region bearing the chemosensory structures. However, the use of visual cues in mating behaviour by snakes is largely unstudied. At least in some species males use visual stimuli when searching for a mate (*e.g.*, *Thamnophis sirtalis*: *Shine et al., 2005*; *Emydocephalus annulatus*: *Shine, 2005*). In addition, sexual dimorphism in eye size has already been reported in numerous snake species (*Faiman et al., 2018*). Thus male-biased sexual dimorphism in the orbital region may be controlled by sexual selection and related to more effective mate searching by males whereas female-biased sexual dimorphism in feeding-related structures may be driven by natural selection (ecologically-based sexual dimorphism).

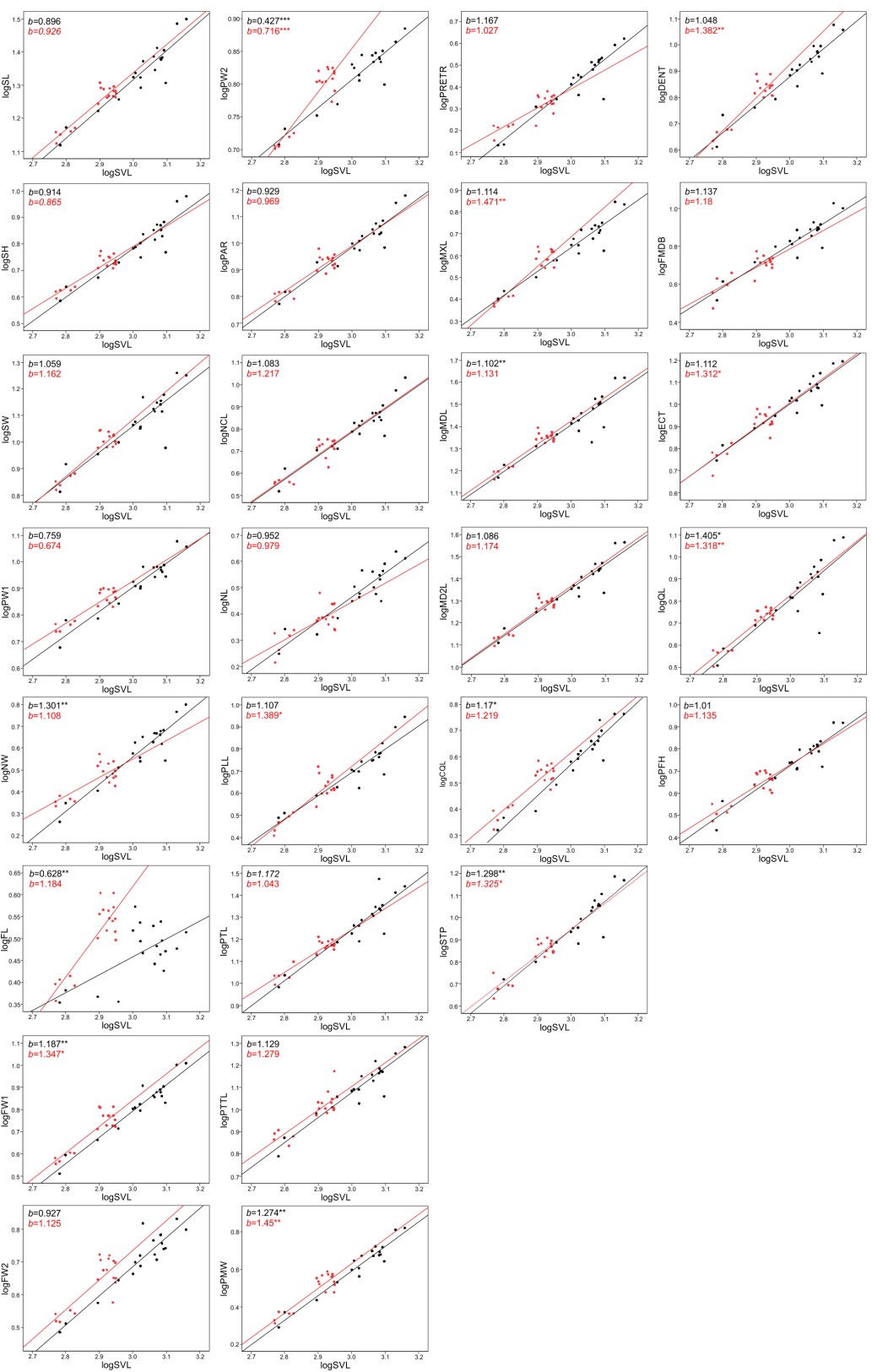

**Figure 4 Patterns of growth of skull dimensions scaled against the snout-vent length (SVL) in the sea krait _Laticauda colubrina_.**

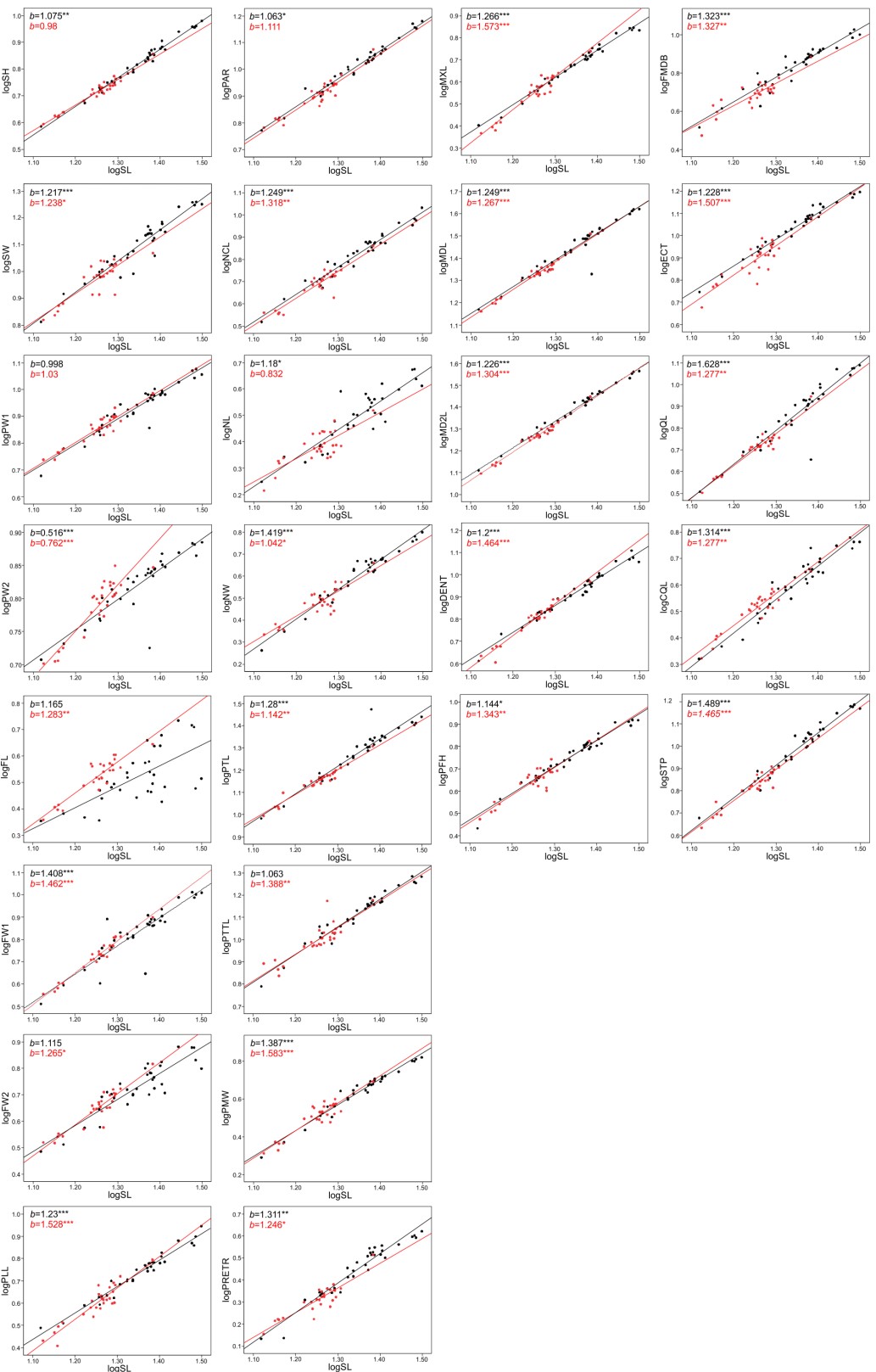

**Figure 5 Patterns of growth of skull dimensions scaled against the skull length (SL) in the sea krait *Laticauda colubrina*.**

## Skull allometry

Skull length, height and width exhibited negative or isometric allometry when compared to SVL, as observed in other snake species (*e.g.*, *Hampton, 2014*; *Hampton & Kalmus, 2014*; *Murta-Fonseca & Fernandes, 2016*; *Jayne et al., 2022*); however, the slopes are remarkably higher in the sea krait compared to the other snakes (Table 3). For example, in the natricine *Nerodia fasciata* (*Hampton, 2014*) and *Farancia abacura* (*Hampton & Kalmus, 2014*), slopes for SL were 0.62 or 0.56 respectively *vs.* 0.9–0.93 in this study (Table 3). Thus, although all species show negative allometry of the head with respect to SVL, in sea kraits the relative skull length is only slowly shrinking with respect to SVL. Moreover, the suspensory elements (QL and STP) and jaw and palate bones (MXL, MDL, PTL, PLL) in *Laticauda* grow with positive allometry with respect to SVL, whereas in other species for which comparable data are available they show negative allometry, with exception for QL in *Boiga irregularis* (Table 3; *Hampton & Moon, 2013*; *Hampton, 2014*; *Hampton & Kalmus, 2014*; *Borczyk et al., 2021*; *Jayne et al., 2022*; *Patterson et al., 2022*). This increase in the growth rate of structures directly influencing gape size and prey swallowing efficiency may underlie the adaptation to piscivory and aquatic habits in *Laticauda*. Specifically, an aquatic lifestyle promotes selection for rapid swallowing of prey and is often reflected by the elongation of suspensory elements that contribute to gape size and increase anteroposterior mandibular excursion during feeding (*Savitzky, 1983*). It is also possible that such a growth pattern is constrained by the space for venom glands. However, to safely evaluate the impact of feeding habits on the allometric patterns of skull bones, taxonomically broader sampling is needed, since some of the differences may reflect phylogenetic, ecological or morphological constraints.

Structures associated with feeding performance, namely the jaws, palatine and pterygoid bones and suspensory elements, scale with positive allometry with the skull length. This is because bigger snakes can usually eat relatively larger prey items and better handle prey items (*e.g.*, *Vincent et al., 2006*). Relative elongation of the structures that influence the gape size (jaws, quadrates and supratemporals) and help with prey handling (maxilla and dentaries) may be a morphological response to increasing prey size. A similar pattern was found in the natricine *Nerodia sipedon*, where a dietary shift from fishes to anuran prey has been observed (*Vincent & Herrel, 2007*), in *Nerodia fasciata* (*Vincent et al., 2007*; *Hampton, 2014*) and in the elapid *Pseudonaja affinis* (*Patterson et al., 2022*). In the elapid *Notechis scutatus* it has been shown that a rapid shift in trophic bone size occurs in response to larger prey (*Ammresh et al., 2023*). In sea kraits this may be explained as an adaptation to a dietary shift from smaller muraenid eels to bigger congerid eels (*Shine et al., 2002*; *Shetty & Shine, 2002*). However, because a similar pattern is also observed in *Aipysurus eydouxii*, a fish-egg eater with no prey-size constraint in feeding morphology (*Borczyk et al., 2021*), it may also simply be a functional and structural consequence of size increase (see *Bonduriansky, 2007*).

Interestingly, with the exception of PW2, all traits that showed sexual differences in growth trajectory are feeding structures. Quadrate length is one of the factors responsible for the maximum gape size, thus limiting the maximum prey diameter that a snake is able to swallow (*e.g.*, *Gripshover & Jane, 2021*). The palato-pterygoid bar plays a role in prey

Borczyk (2023), *PeerJ*, DOI 10.7717/peerj.16266

**Table 3** **Comparison of allometric coefficients (slope) for the sea krait (*Laticauda colubrina*) and different species of colubroid snakes for selected skull characteristics.** Only studies using comparable methods of slope estimation and homologous distances are included. For *L. colubrina* and *A. eydouxii* both male and female data are presented, and for other species data from mixed samples are presented, as the authors of those studies did not conduct separate analyses for the two sexes. Sources: 1 – this study, 2–*Borczyk et al. (2021)*, 3 –*Patterson et al. (2022)*, 4 –*Hampton (2014)*, 5 –*Hampton & Kalmus (2014)*, 6 –*Jayne et al. (2022)*, 7 –*Hampton & Moon (2013)*.

| baseline | | SVL | | | | | | | | | | | SL | | | | | | | | | | |
|---|---|---|---|---|---|---|---|---|---|---|---|---|---|---|---|---|---|---|---|---|---|---|---|
| trait | | SL | PW1 | PL | SW | FL | STP | QL | MXL | PLL | PTL | MDB | PW1 | PL | SW | FL | STP | QL | MXL | PLL | PTL | MDB | |
| *L. colubrina* | F | 0.9 | 0.9 | 0.93 | 1.06 | 0.63 | 1.3 | 1.41 | 1.11 | 1.11 | 1.17 | 1.1 | 1.00 | 1.06 | 1.06 | 1.31 | 1.49 | 1.63 | 1.27 | 1.15 | 1.15 | 1.25 | 1 |
| | M | 0.93 | 0.89 | 0.97 | 1.16 | 1.18 | 1.33 | 1.32 | 1.47 | 1.39 | 1.04 | 1.13 | 1.03 | 1.11 | 1.24 | 1.28 | 1.47 | 1.28 | 1.57 | 1.29 | 1.03 | 1.27 | |
| *Aipysurus eydouxii* | F | 0.66 | | | 0.73 | | | | | | | | 1.12 | 1.03 | 1.3 | 1.43 | 1.25 | 1.7 | 1.87 | 1.72 | 1.36 | 1.28 | 2 |
| | M | 0.79 | | | 1.11 | | | | | | | | 1.14 | 0.97 | 1.34 | 1.16 | 1.62 | 1.64 | 1.57 | 1.45 | 1.12 | 0.96 | |
| *Pseudonaja affinis* | ? | – | – | – | – | – | 0.71 | 0.89 | 0.69 | 0.7 | 0.67 | 0.68 | – | – | – | – | 1.22 | 1.53 | 1.19 | 1.2 | 1.16 | 1.16 | 3 |
| *Nerodia fasciata* | ? | 0.62 | 0.53 | 0.54 | 0.5 | 0.57 | 0.8 | 0.94 | 0.66 | 0.9 | 0.76 | 0.75 | 0.87 | – | 0.82 | – | 1.3 | 1.51 | 1.07 | 1.47 | 1.24 | 1.22 | 4 |
| *Farancia abacura* | ? | 0.56 | 0.58 | – | – | – | 0.75 | 0.89 | 0.63 | 0.8 | 0.72 | 0.68 | 1.03 | – | | – | 1.33 | 1.59 | 1.12 | 1.43 | 1.29 | 1.21 | 5 |
| *Boiga irregularis*\* | ? | 0.67 | – | – | – | – | – | 1.19 | – | – | – | – | – | – | – | – | – | 1.73 | – | – | – | – | 6 |
| *Crotalus atrox* | ? | 0.66 | 0.53 | – | – | – | 0.75 | 0.92 | – | – | – | 0.76 | – | – | – | – | – | – | – | – | – | – | 7 |
| *Python morulus bivittatus*\* | ? | 0.73 | – | – | – | – | – | 0.74 | – | – | – | – | – | – | – | – | – | 1.00 | – | – | – | – | 6 |

transport, as the movements of this complex are responsible for pushing the prey into the oesophagus during the oral and oro-cervical phases of prey transport (the so called 'pterygoid walk'; *Kley & Brainerd, 1999*). Because adult females feed mostly on congerid eels whereas males prey on smaller muraenid ells (*Shine et al., 2002*; *Shetty & Shine, 2002*), these differences in the growth trajectory of the quadrate length (faster in females) may be an adaptation to feeding on bigger prey.

When looking at the skull in general, there was also an interesting pattern of changes in proportions between the anterior-middle-posterior skull segments. The nasal component (NCL) grows with relatively high positive allometry (the slope is equal to ~1.3) and the same is true for frontal length (FL, slope approximately 1.2–1.3), but the parietal component of skull length (PAR) grows almost isometrically (1.0–1.1, see S2). These distances all contribute to the skull length and show that during snake ontogeny the anterior (preorbital and orbital) region grows faster than the posterior one, *i.e.*, bigger kraits are relatively longer-snouted than their smaller companions. This largely fits the general pattern observed among tetrapods (*Emerson & Bramble, 1993*) and snakes seem not to be an exception (*e.g.*, *Murta-Fonseca & Fernandes, 2016*; *Borczyk et al., 2021*; *Patterson et al., 2022*). However, at finer scale, this growth trajectory may again underlie the adaptation to aquatic feeding and locomotion, as it has been shown that aquatic snakes have longer and narrower snouts compared to their terrestrial relatives, which reduces the hydrodynamic drag when striking (*Vincent et al., 2009*; *Segall et al., 2016*; *Silva et al., 2018*). A similar pattern of regional changes in skull proportions was found in semiaquatic *Hydrodynastes gigas* (*Murta-Fonseca & Fernandes, 2016*): the nasal component has been strongly elongated (from 15.5% of SL in small snakes to 33.9% of SL in large ones), whereas the frontal and parietal components have shrunk (ca. 3–5% of SL). This pattern therefore suggests a consistent developmental trend among aquatic and semi-aquatic snakes; however, at this point there is not enough data on terrestrial and aquatic close relatives to fully compare the ontogenetic allometric (slope changes) patterns of skull growths in ecological and phylogenetic framework.

Because of the magnitude of differences (in slope coefficients or skull shape), studies that used combined samples (*e.g.*, *Rossman, 1980*; *Hampton, 2014*; *Hampton & Kalmus, 2014*; *Murta-Fonseca & Fernandes, 2016*), although probably correct at the general level, must be interpreted with caution. Females are not just scaled males (and vice-versa) and even subtle shape differences may be underlain by different allometric trajectories (*e.g.*, *Wilson et al., 2022*) However, the availability of osteological material for such studies is generally limited and does not necessarily correspond with researchers' needs in the context of taxonomic and ecological diversity (*Bell & Mead, 2014*).

## CONCLUSIONS

Sea kraits exhibit clear sexual dimorphism in the skull form that may be explained by intersexual differences in the feeding habits as well as reproductive roles. The overall skull growth pattern resembles the typical pattern observed in other tetrapods. However, when compared to other studied snake species, there are some notable differences in the

pattern of skull growth. Ultimately, until more species representing a broader and more ecologically diverse spectrum are studied in a phylogenetic framework, it will be impossible to infer whether those differences are driven by feeding habits, lifestyle or phylogenetic and morphological constraints.

## ACKNOWLEDGEMENTS

I thank Alan Resetar (Field Museum of Natural History) for making it possible to study specimens in his care as well as his kind reception during my visit to the Field Museum, and Stanisław Bury (Jagiellonian University) and Tomasz Skawiński (University of Wrocław) for helpful comments and discussion. Emma Sherratt and anonymous reviewers made valuable comments and suggestions on the earlier version of the manuscript.

### Funding

This work was supported by the Excellence Initiative–Research University (IDUB) programme for the University of Wroclaw. The funders had no role in study design, data collection and analysis, decision to publish, or preparation of the manuscript.

### Grant Disclosures

The following grant information was disclosed by the author:
The Excellence Initiative –Research University (IDUB) programme for the University of Wroclaw.

### Competing Interests

The author declares that there are no competing interests.

### Author Contributions

- Bartosz Borczyk conceived and designed the experiments, performed the experiments, analyzed the data, prepared figures and/or tables, authored or reviewed drafts of the article, and approved the final draft.

### Data Availability

   The raw measurements are available in the Supplemental Files.

### Supplemental Information

Supplemental information for this article can be found online at http://dx.doi.org/10.7717/peerj.16266#supplemental-information.

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
