# Peer review of "Sexual dimorphism in skull size and shape of Laticauda colubrina (Serpentes: Elapidae)"

_PeerJ, doi:10.7717/peerj.16266_

## Round 0.1 · original submission · Major Revisions

Dear Dr. Borczyk:

Thanks for submitting your manuscript to PeerJ. I have now received two independent reviews of your work, and as you will see, the reviewers raised some concerns about the research. Despite this, these reviewers are (quite) optimistic about your work and the potential impact it will have on research studying elapid sexual dimorphism and skull characteristics for snake systematics. Thus, I encourage you to revise your manuscript, accordingly, taking into account all of the concerns raised by both reviewers.

The reviewers both provided a plethora of constructive criticism, including marked up versions of your manuscript! I think their careful attention to your work, combined with their enthusiasm, should make for a substantially improved manuscript upon revision. I agree with the concerns of the reviewers, and thus feel that their suggestions should be adequately addressed before moving forward.

I look forward to seeing your revision, and thanks again for submitting your work to PeerJ.

Good luck with your revision,

-joe

·

Basic reporting

I have reviewed the manuscript entitled “Sexual dimorphism in skull size and shape of Laticauda colubrina (Serpentes: Elapidae)”. I am pleased to see this research being done, as it addresses a fundamental question in biology – how does morphological diversity arise? The paper is generally well written and structured. References are appropriate and it is a self-contained study in a professional format. My main concern is the length of the methods and results sections, which lack some important details.

An important concept introduced in the Introduction is allometry. However the authors has not mentioned the type of allometry under study in this paper. The data here is from a single species, but it is not explained whether the sampling is across specimens of the age class (e.g., adults), which would be static allometry, or across specimens of a range of age classes (e.g., a growth series from juvenile to adult), which would be ontogenetic allometry. This is important to outline and explain. The papers cited on lines 59 to 63 are mostly ontogenetic allometry at a glance – this needs to be explicitly mentioned.

The abstract is also limited and does not provide any conclusions on the study.

Experimental design

Here the author has taken a system known to have strong sexual size dimorphism (SSD) and asked whether there is sexual shape dimorphism (SShD) and how allometry plays a part in this pattern. There are two possible patterns: size dimorphism results in shape dimorphism, where the shape of one sex is predictable because they are further along the allometric trajectory defined by the other sex. The other pattern is that they differ in shape because they have two distinct allometric trajectories. The first would be indicated by an ANCOVA model of size and sex where sex is significant, and the interaction term is not. The second would have both sex and the interaction of sex and size significant. It is a simple design that gives great power in understanding the evolution of these traits.


AIMS: On lines 78, it is noted that females have longer and wider heads. Thus, the aims could be better written to hit on exactly what is to be studied. i.e., aim 1, there is already known SSD, and we also know that snakes (like all vertebrates) have allometric shape differences in the head, so SShD is expected. Instead, this aim can be “Characterise the shape differences of the skull between male and females”.
The second aim gets at the extending slope or changing slope hypothesis I explained in the previous paragraph, and needs some revision to be explicit here. Suggest “do these differences reflect different allometric growth trajectories of males and females, or is the larger female skull shape a predictable extrapolation along the male allometric trajectory?”. The third aim I believe relates to the specific skull measurements and how each changes with size (using the reduced major axis regression analyses). I also suggest a small edit here, to be explicit that you are discussing the male and female separately: “what are the
general allometric patterns of skull growth in males and females of the studied species?”. However this pretty much overlaps with aim 1, in that it is characterising the differences in shape between the two sexes. So therefor, consider two clear aims – characterise the shape variation of the skull. And characterise the difference in allometry trajectories.

The sea krait studied here is fascinating because the two sexes have different diets, and therefore niche separation is the possible cause of SSD. The author has collected an ample dataset of sufficient numbers of males and females from which to test their hypotheses. There is enough power to do this analysis, with 31 females and 26 males. Methodologically, however, the Methods section is very short and lacking in detail. What software was used to do the analyses? The analyses also fall short in a few key places, which I shall outline:

1) There are no graphical displays of the data. It is imperative to graph data in allometry studies (e.g., Packard 2016, The essential role for graphs in allometric analysis, Biological Journal of the Linnean Society). It allows the authors to know what the patterns underlying the data are, and it helps the reader understand the variation. A table of allometric coefficients is important too, but insufficient on its own.

2) In order to study shape as a separate entity of size, prior to examining patterns of allometry, it is necessary to scale the data to be standardised. The common approach to do this is the log-shape ratio (see Mosimann & James 1979 Evolution, and a nice review by Claude 2013 Hystrix). The resulting log-shape ratios are variables for any multivariate analysis. I have demonstrated this in the attached anlaysis code.

3) The paper says a PCA was done, and since scaling was not adjusted for, PC1 is basically all size. This is not so useful for a reader, and I suggest it is deleted. Instead, a PCA of the log-shape ratios could be done to demonstrate the shape differences between the males and females. However, these data are from an allometric series, so it would be more appropriate to examine it with a multivariate regression (see point below).

4) I wanted to see the data plotted, so I took the provided raw data and plotted a multivariate regression of the log-shape ratio. Very exciting! It is super clear that females have a different allometric slope to males. So the second scenario I outlined is supported by the data. This is a very exciting result, and one that is hidden in the current version of this manuscript.

5) The RMA for how each bone varies with size is useful to compare among other studies, and satisfies aim 3. However Table 4 is just not accessible as a summary table for the reader. The pattern is lost in the numbers. I suggest a creative summary of the results, i.e. colour that highlights positive, isometry or negative allometry when it differs between sexes.

6) The MANCOVA on the raw data is not appropriate because all it tells you is that the bones differ in size, which you know because there is SSD. This is why the log-shape ratio method is more appropriate. Therefore I suggest Table 2 is updated with results using log-shape ratios instead.

7) SVL versus head size. This is a very important aspect to consider, as it asks completely different questions. SVL is body size, which differs between the sexes. But asking whether any of the skull measurements vary with body length asks “does the female differ from the male in body shape proportions?” i.e. a smaller head or bigger head on the body. Using skull size, the question then becomes about head size dimorphism and the related skull shape dimorphisms, irrespective of the proportions of the body. I undertook the analyses about with head size because it answers better the question the author posed in the aims. The SVL analysis asks a different question (which is also important, and maybe should be a separate explicit aim). The paper should be clear that these two are separate patterns that covey a different meaning of dimorphism.

Validity of the findings

The author finds sexual dimorphism in many aspects of the skull in the sea krait. The main concern is that sexual shape dimorphism is discussed, but not explicitly tested for, because the data are size variables not shape variables. The biological differences are very clear, but teasing out the differences in absolute size of the bones and the relative size of them is important for the study to be valid. This is easy to do using the log-shape ratio approach, and I’ve provided the analytics for the author.

This study is novel and I’m very excited to see this work done, because it is an overlooked area as noted by the author. Many studies have looked at skull shape allometry but have not considered the differences between males and females. This study reveals a critically important result – the larger females are NOT just large males. And this agrees with the finding of a very prominent paper that has just been published in Nature Communications (Laura B. Wilson et al. 2022). I suggest citing this and leveraging that in the abstract and the discussion.

Additional comments

A few typographical errors in the references (names with extra letters e;g. Shhine, needs fixing.

Reviewer 2 ·

Basic reporting

Overall, this article broadly meets PeerJ’s basic reporting standards, although as indicated below I have a few comments and suggestions for improvements.

In terms of grammar etc, the article is well-written and the main ideas are presented clearly. I’ve submitted an annotated PDF alongside this review, containing a number of minor corrections and suggested edits; I think these would further improve the quality of the manuscript, but the author is ultimately free to incorporate as many or as few of these as he sees fit.

In terms of whether the article included appropriate context for this study (i.e., relevant literature references and sufficient overall background information), again I think that the manuscript is generally successful in this regard. My only comment here is that there are a few recent papers that also discuss snake skull allometry but weren’t cited in this article. I’ve listed these at the end of this section; obviously there’s no obligation to include them, but I just wanted to point them out since I think they might be useful/relevant to incorporate.

Regarding article structure, the manuscript conforms to PeerJ’s suggested format, includes key raw measurement data as a supplementary spreadsheet, and presents all relevant results. As a minor edit, the supplementary data file should be referred to somewhere in-text; for example, adding a reference to “Data S1” in the first paragraph of the Materials & Methods section would address this. My main comments here concern the figure and tables:
→ In terms of showing how the various measurements compare to each other, I think that it would be more effective to move Table 1 (the PCA results) to a Supplementary Information file and replace it in the main text with a figure plotting these results; in my opinion at least, a plot would be a lot easier/faster to interpret than the same information presented in a table.
→ For Figure 1, I’d recommend colour-coding the measurement labels (e.g., use different colours for the labels associated with feeding-related structures vs chemosensory structures vs both feeding & chemosensory vs overall skull form/shape). This would provide a useful visual representation of the classifications/categories described in Paragraph 2 of the Materials & Methods section.
→ For all display items, I think it’d be more convenient to define all abbreviations in each of the respective captions, so that the reader doesn’t have to flip back and forth between the tables/figure and the main text. (Or, if that takes up too much space, listing/defining the abbreviations in alphabetical order in just Figure 1’s caption would also be sufficient.)

Additional relevant papers:
Palci et al. (2016). Patterns of postnatal ontogeny of the skull and lower jaw of snakes as revealed by micro-CT scan data and three-dimensional geometric morphometrics. Journal of Anatomy 229:723–754. https://doi.org/10.1111/joa.12509
Jayne et al. (2022). Scaling relationships of maximal gape in two species of large invasive snakes, brown treesnakes and Burmese pythons, and implications for maximal prey size. Integrative Organismal Biology 4:obac033. https://doi.org/10.1093/iob/obac033
Ammresh et al. (2023). Island tiger snakes (Notechis scutatus) gain a ‘head start’ in life: how both phenotypic plasticity and evolution underlie skull shape differences. Evolutionary Biology 00:1–16. https://doi.org/10.1007/s11692-022-09591-z

Experimental design

The research questions underlying this study are clearly stated, and the associated gap in knowledge is evident and generally well-contextualized in terms of existing research (although see “Basic Reporting” for suggestions of a few other papers to consider including).

Regarding the technical rigour and replicability of this study, my main comments here concern the statistical analyses. Specifically, there are several discrepancies between the Materials & Methods section vs the Results section that, although generally minor, make it difficult to fully assess/understand the statistical approach that was taken herein (and that therefore preclude confident replication of these analyses). For example, the 1st sentence of the Results section refers to the results of an ANOVA test, but this test is not mentioned in the Materials & Methods. Similarly, the Materials & Methods section states that an ANCOVA was performed, even though the Results and Table 2 concern MANCOVA results. And, the Materials & Methods section lists SL, SW, and SH as dependent variables with SVL as the covariate, whereas in Table 2 SL is the covariate and all other skull measurements are dependent variables. From the perspective of a hypothetical future reader of this article, these inconsistencies make it confusing to follow which statistical analyses were done and why (and would therefore make it very difficult to confidently replicate); from the perspective of a reviewer, it’s difficult to assess the validity and rigour of the article’s methodology since these tests aren’t clearly explained. Overall, I therefore can’t really make a confident judgment on whether the statistics are correct/appropriate (and, ultimately, can’t yet recommend this article for publication) until this issue is corrected.

Finally, a few additional comments/suggestions regarding the experimental design:
→ For a few of the measurements, there are discrepancies between the definitions provided in the main text versus the depiction of these measurements in Figure 1. Specifically, the main text states that Parietal Width 2 (PW2) was measured at the widest point of the parietals, but Figure 1 depicts PW2 at the narrowest point; which one of these is correct? Similarly, the main text states that Frontal Length (FL) was measured between the most rostral and most caudal edges of this bone, but Figure 1 depicts FL as terminating anteriorly at the most rostral extent of the interfrontal midline (which is slightly posterior to the most rostral edge of the overall frontal bone). This difference is subtle, but necessary to accurately replicate this measurement; as such, either the main text should be modified to state that FL was measured along the midline, or Figure 1 should be modified to correspond to the definition in the main text.
→ The Abstract states that 61 snakes were studied, but the Materials & Methods section says 57 skulls and the supplementary data file includes 60 specimens. Please double-check the sample size and correct/clarify the main text as required.
→ In addition to adding a figure for the existing PCA data (see comment in “Basic Reporting” section), I’d also suggest running another PCA to determine how the specimens plot out in morphospace (in other words, flipping the rows/columns from the original PCA). Since one of this study’s main findings is that many skull elements are dimorphic in females vs males, this morphospace plot might be useful in graphically demonstrating those differences, as well as revealing which measurements drive the observed morphospace occupation.
→ Given that several important feeding-related muscles attach to the retroarticular process of the mandible, I would also consider the length of this process (PRETR herein) to be a feeding-related structure; this bone should either be included when discussing feeding-related adaptations etc later on in the manuscript, or an explanation should be provided in the Materials & Methods section as to why it wasn’t considered as such.

Validity of the findings

Regarding overall validity of the article’s findings, my comment above about clarifying the statistical approaches is also relevant here. However, apart from this issue, I found the Discussion section of this article to be interesting, well-written, thorough in its analysis of the results recovered herein, and well-reasoned in terms of the ecological/adaptive explanations posited for these results. My only suggestion regarding the Discussion is to slightly rearrange its “Skull allometry” subsection; specifically, paragraph 3 and the latter part of paragraph 1 of this subsection both concern allometric scaling of the jaw bones, so I’d suggest editing this section to discuss this topic all in one spot.

Additional comments

In this article, the author uses linear morphometric measurements of male and female sea krait skulls to test for the presence of sexual dimorphism. Although overall skull size and shape were not found to differ between males vs females, specific skull bones did show key differences in size and allometric trajectory. Specifically, feeding-related bones tended to be larger and follow a steeper allometric slope in females, whereas bones in the anterior skull tended to be larger in males. Returning to his initial hypotheses / research questions, the author discusses these results as reflecting adaptations for diet and mate-searching, respectively. Comparisons of allometric trends in sea kraits vs other snakes also reveal the potential influence of aquatic adaptations on the skull of Laticauda.

As stated elsewhere in this review, I found this article to be well-written and interesting, although certain aspects of the statistical methods should be clarified prior to being accepted for publication. Below, I’ve summarized the comments/suggestions made in the other sections of this review and categorized them as “Essential Revisions”, “Major Comments (highly recommended but not mandatory)”, and “Minor Comments (summary from above)”. I’ve also included 2 subsections of additional minor comments that didn’t fall under the other sections of this review form. Within each category below, comments are listed in order of importance.

Essential Revisions:
1. As explained in more detail under the “Experimental Design” section of this review, the Materials & Methods and Results sections of the manuscript should be thoroughly proofread for accuracy and consistency. In my opinion this is a mandatory revision, as it’s currently rather confusing to follow which statistical analyses were performed, why, and whether they were performed accurately. Because of this, I can’t fully recommend this manuscript for publication in its current state. However, this is the only major issue that I have with this article; once it’s addressed (which should be fairly straightforward to do), I think this study will be a useful and interesting contribution to the literature on snake skull anatomy and evolution.
2. Check for consistency in how PW2 & FL are described in the Materials & Methods section versus how they’re depicted in Figure 1 (see full comment under “Experimental Design”).
3. Double-check sample size (see full comment under “Experimental Design”).

Major Comments (highly recommended but not mandatory):
1. Add a figure for the existing PCA data, and move Table 1 to a supplementary file (see full comment under “Basic Reporting”).
2. Perform an additional PCA to show the distribution of the observed specimens across morphospace (see full comment under “Experimental Design”).
3. Take a look at suggested papers (see “Basic Reporting” section: Palci et al. 2016; Jayne et al. 2022; Ammresh et al. 2023) and incorporate into Introduction & Discussion if useful.
4. Consider including the length of the retroarticular process of the mandible (PRETR) in discussions/comparisons of feeding-related elements (see full comment under “Experimental Design”).

Minor comments (summary from above):
1. Correct minor typos and grammatical corrections (see annotations in accompanying PDF).
2. Colour-code the labels in Figure 1 (see full comment under “Basic Reporting”).
3. Consider re-arranging the “Skull allometry” subsection of the Discussion (see comment under “Validity of the Findings”).
4. Add a reference to the supplementary data file in the main text (see “Basic Reporting” section).
5. Include abbreviations in figure & table captions (see full comment under “Basic Reporting”).

Additional minor comments (Methods):
1. Is there any data available regarding the age or ontogenetic stage of the specimens included in this study? If so, this would be useful to indicate in the first few sentences of the Materials & Methods section (e.g., by adding a sentence like: “Of these specimens, __ females and __ males were adults, __ females and __ males were juveniles, and the remaining specimens were of unknown ontogenetic stage.”). If not, then stating the range of body sizes among males and females would be helpful to give the reader some overall context for the dataset underlying the paper (e.g., a sentence such as: “Female specimens ranged from ___ to ___ mm in length, and male specimens ranged from __ to ___.”).

Additional minor comments (Edits to wording etc):
1. Usage of abbreviations throughout the manuscript should be double-checked for consistency. For example, Tables 1 and 2 (and the raw data spreadsheet) include “PFL” and “PML”, but these aren’t described in the main text or included in Figure 1 or the other tables. Similarly, “MDB” and “MDBL” are each used once in the Discussion, but are presumably typos since these abbreviations aren’t used or defined elsewhere. And finally, “EMBD” is included in the list of feeding-related structures in the Materials & Methods, but this isn’t defined in the previous paragraph or in Figure 1 (I assume this should be “FMDB” / mandibular fossa length).
2. The Abstract states that “The sea krait Laticauda colubrina (Serpentes: Elapidae) exhibits very clear sexual dimorphism in body size, with females being bigger and having a relatively longer and wider head.” However, I’d recommend modifying the latter part of that sentence to something like: “…with previous studies having reported females to be larger and to have a relatively longer and wider head”. In this case, the present manuscript found that there *wasn’t* a significant difference in overall skull length and width for males vs females, so I think it’s important to be really clear here that this sentence is referring to previous literature. I have a similar recommendation for Lines 77–79 (toward the end of the Introduction), where it states that “This snake species exhibits very clear sexual dimorphism in body size, with females being bigger and having relatively longer and wider heads (Shetty & Shine, 2002; Shine et al. 2002).” Again, I would modify this sentence, perhaps to something like: “Previous studies have found this species to exhibit very clear sexual dimorphism in body size, with females being…”.
3. This is quite minor, but “skull length (SL)” should be included in the list of measurements in Paragraph 1 of the Materials & Methods section (it’s defined in a later paragraph, but would make more sense to include here, with the rest of the measurement definitions).
4. Quadrate length (QL) is discussed in the context of feeding-related adaptations several times in the Discussion, but isn’t included in the list of feeding-related structures in Paragraph 2 of the Materials & Methods section (I assume this measurement was just accidentally left out of this list).
5. At the beginning of Paragraph 2 of the Discussion, I would add a sentence such as: “Although females and males did not differ in overall skull shape (as reflected by SL, SH, and SW), specific bones within the skull did show significant sexual dimorphism.” As currently written, the transition between Paragraphs 1 and 2 in the Discussion is a bit abrupt, so I think this addition would be useful in clarifying that the previous paragraph concerned the overall skull dimensions whereas the current paragraph concerns differences among specific bones.

Annotated reviews are not available for download in order to protect the identity of reviewers who chose to remain anonymous.

---

## Round 0.2 · Minor Revisions

Dear Dr. Borczyk:

Thanks for revising your manuscript. The reviewers are mostly satisfied with your revision (as am I). Great! However, there are edits to make and a few more issues to address. Please tend to these matters these ASAP so we may move towards acceptance of your work.

Best,

-joe

·

Basic reporting

The revised manuscript is much improved, and the new figures are integral additions to showing the results of this study.
The abstract is more informative and the discussion much improved also.
The author has revised the manuscript following several of the suggestions of the two reviewers, however some of the revisions are incomplete and there are some further revisions to make (detailed below)

Experimental design

New aims are good and clearly inform the reader. Details of the specimens have been added.

Validity of the findings

New figures are good and better represent the data. Discussion is well written and clearly outlines the findings and places them in context of the literature.

Additional comments

Reviewer 2 suggested 3 relevant papers that assess allometry in snake skulls, which are relevant to the second paragraph of the Introduction. The authors says “Thank you for pointing this, I added these papers”. However the manuscript does not show these revisions (paragraph starting Line 62). Ammresh et al. is added to the reference list but not cited in the manuscript. Palci et al is not included anywhere. Jayne et al. added to the discussion only. These are papers that satisfy the sentence in Line 62 and should be included here.

Given that in references are being grouped by family in this paragraph, is is necessary to give the elapid families being referred to on lines 66-68.

Also there is a missing semicolon after Dipsadidae.

The families should all be written consistently, so either all in capitalised full name, or lower case common usage name. Currently it is a mixture.

Line 152 geomorph with small g

Throughout the methods, this section should be written in past tense. So “I run” needs to be changed to “I ran”.

The abstract results section could also be revised now the new analyses find more specific differences so that here it can be given instead of a vague “the sexes differ in shape”. Males and females have distinct allometric trajectories, and males are not simply scaled up females.

As remarked by myself and Reviewer 2 previously, the number of tables in the main manuscript is likely excessive as there are 6, three of which are a whole page in size. Tables 1, 4 and 5 are large and not immediately accessible to a reader. Table 1 should be supplementary, and the important loadings and take home information reported in the results section as clear summary sentences. Tables 4 and 5 this information can be included with each part of figure 3 and 4. That way the reader is not jumping between graph and table for the coefficients and significance values. The current Results section has very little to say about the main findings from these tables, which means the reader has to do a lot of work to interpret them. But if they are not very important, then all the more reason to place them as supplementary.

Similarly the two new figures of all scatterplots are large and the text is very small at full size on a A4 page. Can the most important variable be provided on the main figure, and the remaining be in supplementary? Otherwise these figures are very time consuming to interpret and are mostly not important to the narrative.

Reviewer 2 ·

Basic reporting

No comment.

Experimental design

No comment.

Validity of the findings

No comment.

Additional comments

Overall, I think that the author has successfully addressed/incorporated the main comments provided in the first round of reviews. In particular, the manuscript has really been strengthened by re-analysing the data based on the other reviewer’s comments, as well as the addition of colour-coding to some of the tables / figures. My major comments from the initial submission concerned the clarity of the Methods & Results sections, which again I think have been addressed quite well in this revision.

I do have a few remaining suggestions to further improve the paper, as listed below; however, these are all ultimately optional, as I do think that the manuscript would be suitable for publication in its present form.

[1] As also noted in the first round of reviews, I’d again encourage the author to present the PCA results as a figure in addition to Table 1. The table is certainly useful in seeing how much each measurement contributes to overall variation in the dataset, but doesn’t convey how the specimens themselves compare to each other, which would be helpful in this kind of analysis. (Again, though, this is optional.)

[2] In Table 1, the value of QL for PC1 should also be bolded (since it’s > |0.7|). Also, the value of PFL is bolded under PC2, but this variable and PC aren’t mentioned in the Results / Discussion – perhaps either un-bold this value, or add a short sentence to the Results mentioning this contribution to PC2.

[3] The caption for Figure 2 is incomplete (looks like maybe a formatting issue with inputting the caption during article submission?). Anyway, verify the caption, and provide an explanation/legend for the colours.

[4] Throughout the figures & tables, consider using less-repetitive colour schemes -- for example, red & blue reflect feeding structures vs sensory structures in Figure 1, whereas in Tables 4–5 they reflect positive vs negative allometry. This is a bit confusing at first glance (although the captions do explain these meanings), so I’d suggest only using red/blue for one of these and choosing different colours for the rest.

[5] There are a few minor grammatical errors to correct (see annotated PDF attached to this review).

Annotated reviews are not available for download in order to protect the identity of reviewers who chose to remain anonymous.

---

## Round 0.3 · accepted · Accept

Dear Dr. Borczyk:

Thanks for revising your manuscript based on the concerns raised by the reviewers. I now believe that your manuscript is suitable for publication. Congratulations! I look forward to seeing this work in print, and I anticipate it being an important resource for groups studying elapid sexual dimorphism and skull characteristics for snake systematics.

Thanks again for choosing PeerJ to publish such important work.

Best,

-joe